# New predictive model of the touchdown times in a high level 110 m hurdles

**Ryo Iwasaki**[1,2]*, **Hiroyuki Nunome**[1], **Kento Nozawa**[3]

**1** Faculty of Sports and Health Science, Fukuoka University, Fukuoka, Japan, **2** United Graduate School of Education, Tokyo Gakugei University, Tokyo, Japan, **3** Graduate School of Frontier Science, The University of Tokyo, Chiba, Japan

\* r-iwasaki@fukuoka-u.ac.jp

**Data Availability Statement:** https://github.com/iwasaki71/race_predict.

**Funding:** The authors received no specific funding for this work.

## Abstract

The present study aimed to establish a more robust, reliable statistical model of touchdown times based on the data of elite 110 m hurdlers to precisely predict performance based on touchdown times. We obtained 151 data (race time: 13.65 ± 0.33 s, range of race time: 12.91 s– 14.47 s) from several previous studies. Regression equations were developed to predict each touchdown time (times from the start signal to the instants of the leading leg landing after clearing 1st to 10th hurdles) from the race time. To avoid overtraining for each regression equation, data were split into training and testing data sets in accordance with a leave–one–out cross-validation. From the results of cross-validation, the agreement and generalization were compared between the present study model and the existing model. As a result, the proposed predictive equations showed a good agreement and generalization ($R^2$ = 0.527–0.981, $MSE$ = 0.0015–0.0028, $MAE$ = 0.019–0.033) compared to that of existing equations ($R^2$ = 0.481–0.979, $MSE$ = 0.0017–0.0039, $MAE$ = 0.034–0.063). Therefore, it can be assumed that the proposed predictive equations are a more robust, reliable model than the existing model. The touchdown times needed for coaches and elite hurdlers to set their target records will be accurately understood using the model of this study. Therefore, this study model would help to improve training interventions and race evaluations.

## Introduction

In the 110 m hurdles, ten hurdles of 1.067 m in height are evenly placed at 9.14 m intervals along a straight course of 110 m. Since the height of the hurdle for 110 m hurdles is substantially higher than that of the 100 m hurdles (0.838 m), the 110 m hurdles face higher technical demands than the 100 m hurdles for women [1, 2]. Thus, 110 m hurdlers are required to achieve both high sprint velocity and high-level hurdle clearance techniques by minimizing the reduction of horizontal velocity during hurdle clearance throughout the race [3–5]. It has been reported for elite hurdlers that the interval times between hurdles were strongly correlated (r = 0.81–0.99; $p < 0.05$) with the resultant race performance [5]. Another previous study also showed that faster hurdlers run at a higher step frequency during the interval running (except the landing step) rather than slower hurdlers do [6]. Theoretically, acquiring a higher

**Competing interests:** The authors have declared that no competing interests exist.

step frequency is an essential element to reduce interval times because the number of steps to cover the interval distance between hurdles is predetermined, comprising three steps [7, 8].

Spatiotemporal variables have been recognized as important determinants of race performances in sprint and hurdle events [8–11] and also as modifiable parameters through training sessions [9]. In the hurdle events, times from the start signal to the instants of the leading leg landing after clearing the 1st to 10th hurdles (touchdown time) are typical parameters. Thus, touchdown times of elite hurdlers would provide coaches and practitioners with supportive information for a better understanding of performance evaluation and more effective training. This is necessary to precisely estimate touchdown times in a robust manner.

Notation analysis has been used to identify the characteristics of elite race performance in sprint events, and several attempts have been made for the 110 m hurdles [5, 6, 12–14]. From these studies, touchdown times and interval times were used to assess the resultant race performance. Coaches often use these temporal parameters in their daily training sessions as a target of training [15]. Tsiokanos et al. [5] claimed the necessity of statistical models that provide temporal parameters to make effective training interventions.

Miyashiro et al. [15] established a predictive model of touchdown time in 110 m hurdles. To date, the model appears to be the only one based on a statistical analysis of actual races of the 110 m hurdles. However, in fact, the model was established from the data of intermediate-level hurdlers (their time ranged from 13.71 s to 14.59 s), their range of records was substantially slower than the mean record (13.50) of the world 50th ranking during the last decade (2010 to 2021). The existing model, therefore, fails to reliably predict the touchdown times of elite hurdlers. Miyashiro et al. [15] evaluated the prediction accuracy of the regression model using the training data. Training data in this context refers to the data used to develop the regression model. Clearly, the training data should be removed from the dataset to assess the regression model, which has been known as an essential procedure to give the model more veracity and generalization to predict unseen races [16]. Otherwise, the overfitted model [17] likely provides inaccurate information for coaches who might instruct their hurdlers in an inappropriate way. Based on the previous discussion, it is clear that the existing model does not provide accurate predictions. Therefore, a more robust, reliable model established in an appropriate manner, such as LOOCV [18], is required. The present study, therefore, aimed to establish a more robust, reliable statistical model of touchdown times based on the data of elite 110 m hurdlers to precisely predict their typical touchdown times. This model would provide useful information for coaches who train sub-elite or intermediate-level hurdlers. We hypothesized that the model established in this study would show better generalization performance compared to the model of Miyashiro et al. [15].

## Materials and methods

### Data collection

We obtained 154 data samples from seven previous studies [5, 13, 14, 19, 20]. These data were obtained from competitions. A few individual data were excluded from the analysis if the touchdown times drastically changed (more than 0.10 s) during a single race. For instance, in the 2004 Olympic Games, the run-in section time (from the 10th hurdle landing to the finish) of the 8th placed hurdler drastically increased due to hitting the 10th hurdle. Consequently, 151 data samples, including the Olympic Games and the International Association of Athletics Federations (IAAF) World Championships, were analyzed (mean ± SD, race time: 13.65 ± 0.33 s, range: 12.91 s– 14.47 s).

We performed a power analysis using G*power [21] for each variable (see below) and confirmed the statistical power $(1 - \beta)$ of $> 0.90$.

## New predictive equations proposal and evaluation metrics

The Shapiro-Wilk test was performed to ensure the data distribution normality for race time and each touchdown time (EZR ver.1.54, Saitama Medical Center, Jichi Medical University, Japan; Kanda [22]). After the data normality was confirmed (Shapiro–Wilk test, $p > 0.05$), regression equations were developed to estimate each touchdown time from the race time with a 95% confidence interval and 95% prediction interval.

To avoid overtraining for each regression equation, data were split into training and testing data sets in accordance with a leave–one–out cross-validation (*LOOCV* [23]). *LOOCV* is a particular case of $k$–fold cross–validation when $k = 1$. Given $N$ data points, *LOOCV* derives $N$ training and testing sets. Each testing set contains only one data point once, and the training sets contain the rest of the data points. We trained regression equations on each training set and evaluated the trained regression equations on the corresponding test set. Since we have $N$ regression equations per touchdown, we report the averaged equation to show the trained regression models.

For each equation in the present study and Miyashiro et al. [15], the goodness of fit was estimated with the adjusted coefficient of determination ($R^2$) of the linear regression and the mean squared error (*MSE*) from *LOOCV*. The *MSE* was calculated as follows:

$$MSE = \frac{1}{N} \sum_{i=1}^{N} ||y_i - \hat{y}_i||_2$$

where $N$ is the number of subjects, $y_i$ is the actual value of $i$ th subject and $\hat{y}_i$ is the predictive value from $i$ th subject's race time, respectively. Similarly, to compare the generalization performance of the two models to the elite hurdlers, the mean absolute errors (*MAE*) with the actual data were calculated for all touchdown times. The *MAE* was calculated as follows:

$$MAE = \frac{1}{N} \sum_{i=1}^{N} |y_i - \hat{y}_i|$$

The test data for *MAE* was the race records corresponding to the world 50th rankings (mean record of the world 50th ranking during the last decade) were extracted as test data from the previous study [24] ($N = 9$). The predictive touchdown times were calculated by substituting these data into the two regression models (the new proposed model from the *LOOCV* and the model of Miyashiro et al. [15]). In addition, to examine the impact of reaction time on predictive performance, a numerical experiment for net touchdown times (without the reaction time) was completed. The net touchdown times with reported reaction times were extracted from the data of touchdown times ($N = 81$). In this numerical experiment, *MAE* was not calculated because some test data for *MAE* did not report reaction time. Moreover, for *MSE*, although the magnitude of *MSE* should be evaluated against the variable scale [25], TD times during the hurdle race monotonically accumulate from TD1 to TD10, which induces a substantial increase in their scales towards the end of the race. To examine the effect of increasing the scale of TD times on their *MSE*, we conducted the following numerical experiment for each touchdown time to unify the scale. We applied z-score standardization to each touchdown time as a pre-process before fitting the regression models:

$$z(i, t) = \frac{y_i - y_{avg(t)}}{std(t)}$$

where $i$ is the index of data samples (i.e., from 1 to 151), $t$ is each touchdown position (i.e., from TD1 to TD10), $z(i, t)$ is the scaled touchdown time of $i$, $std(t)$ is the standard deviation

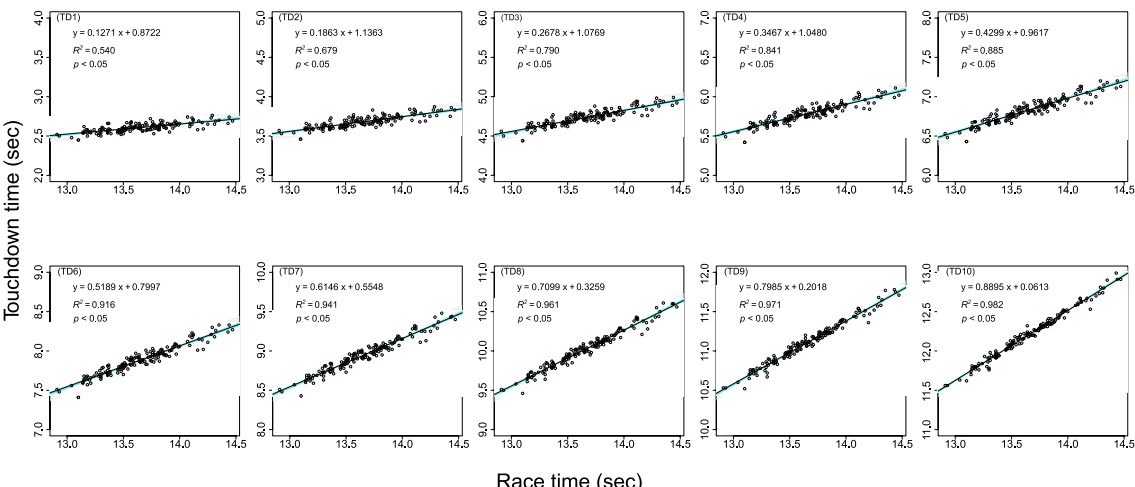

**Fig 1. Relationships between the race time and each touchdown time (TD).** Shaded areas in blue indicate the 95% confidence interval for regression lines. Shaded areas in gray indicate the 95% prediction interval for estimated touchdown times.

of $t$, $y_i$ is the actual touchdown time of $i$ and $y_{avg\ (t)}$ is the average touchdown time at $t$, respectively. Note that this numerical experiment was carried out only on the present study model because the training data used for the model of Miyashiro et al. [15] was not available. $R^2$ and $MSE$ were calculated using the $z\ (i,\ t)$. These processes were implemented with a Python library, scikit-learn [26].

To compare these evaluation metrics between the present study and the model of Miyashiro et al. [15], Student's $t$-tests were used. Statistical significance was set at $p < 0.05$.

## Results

As shown in Fig 1, a significant regression equation was obtained at each touchdown time against the resultant race time ($p < 0.05$).

Table 1 compares the goodness of fit and generalization in these regression equations between the model of the present study and the model of Miyashiro et al. [15]. Overall, the

**Table 1. Comparison of goodness of fit and generalization in each equation between the present study model and Miyashiro model.**

| Touchdown | $R^2$ | | $MSE$ | | $MAE$ | |
|---|---|---|---|---|---|---|
| | Present study | Miyashiro et al. | Present study | Miyashiro et al. | Present study | Miyashiro et al. |
| TD1 | **0.527** | 0.481 | **0.0015** | 0.0017 | **0.019** | 0.034 |
| TD2 | **0.670** | 0.595 | **0.0018** | 0.0022 | **0.028** | 0.051 |
| TD3 | **0.784** | 0.722 | **0.0021** | 0.0027 | **0.026** | 0.054 |
| TD4 | **0.836** | 0.778 | **0.0025** | 0.0034 | **0.029** | 0.063 |
| TD5 | **0.881** | 0.840 | **0.0027** | 0.0036 | **0.032** | 0.063 |
| TD6 | **0.914** | 0.878 | **0.0028** | 0.0039 | **0.029** | 0.054 |
| TD7 | **0.939** | 0.917 | **0.0027** | 0.0036 | **0.028** | 0.049 |
| TD8 | **0.959** | 0.947 | **0.0023** | 0.0030 | **0.033** | 0.041 |
| TD9 | **0.970** | 0.962 | **0.0022** | 0.0027 | **0.031** | 0.038 |
| TD10 | **0.981** | 0.979 | **0.0017** | 0.0019 | **0.028** | 0.031 |
| Mean (SD) | **0.846 (0.141)** | 0.810 (0.160) | *__0.0022 (0.0004)__ | *0.0029 (0.0007) | *__0.028 (0.004)__ | *0.048 (0.011) |

TD: touchdown time. $R^2$: adjusted coefficient of determination. $MSE$: mean squared error. $MAE$: mean absolute error.

*: Significant difference between the present study and the model of Miyashiro et al. ($p < 0.05$).

**Table 2. Comparison of regression equations with and without reaction time.**

| Touchdown | Reaction time | Equation | F value (1, 79) | $R^2$ | MSE |
|---|---|---|---|---|---|
| TD1 | With | 0.1322 + 0.6543 | 66.9* | 0.452 | 0.0017 |
| | Without | 0.1330 + 0.7922 | 78.7* | **0.493** | **0.0015** |
| TD2 | With | 0.1879 + 0.8184 | 140.0* | **0.635** | **0.0017** |
| | Without | 0.1887 + 0.9562 | 106.3* | 0.568 | 0.0022 |
| TD3 | With | 0.2695 + 0.7574 | 239.8* | **0.749** | **0.0020** |
| | Without | 0.2703 + 0.8953 | 183.2* | 0.695 | 0.0026 |
| TD4 | With | 0.3488 + 0.7225 | 310.5* | **0.795** | **0.0026** |
| | Without | 0.3496 + 0.8604 | 242.2* | 0.751 | 0.0033 |
| TD5 | With | 0.4320 + 0.6340 | 460.0* | **0.852** | **0.0027** |
| | Without | 0.4328 + 0.7718 | 349.5* | 0.813 | 0.0035 |
| TD6 | With | 0.5215 + 0.4653 | 664.5* | **0.892** | **0.0027** |
| | Without | 0.5223 + 0.6031 | 496.2* | 0.861 | 0.0036 |
| TD7 | With | 0.6129 + 0.2776 | 1012.0* | **0.927** | **0.0024** |
| | Without | 0.6122 + 0.4368 | 734.1* | 0.902 | 0.0034 |
| TD8 | With | 0.7088 + 0.0413 | 1515.0* | **0.950** | **0.0022** |
| | Without | 0.7096 + 0.1792 | 1036.0* | 0.902 | 0.0032 |
| TD9 | With | 0.7965–0.0726 | 2121.0* | **0.964** | **0.0020** |
| | Without | 0.7973 + 0.0652 | 1406.0* | 0.946 | 0.0030 |
| TD10 | With | 0.8899 + 0.2452 | 3350.0* | **0.977** | **0.0016** |
| | Without | 0.8907 + 0.1074 | 2042.0* | 0.962 | 0.0025 |

TD: touchdown time. $R^2$: adjusted coefficient of determination.

MSE: mean squared error.

*: $p < 0.05$

model of the present study yielded a higher adjusted coefficient of determinations ($R^2$) and lower generalization indexes (MSE, MAE) compared to those of the model of Miyashiro et al. [15]. Of these variables, the mean values of MSE ($p = 0.035$) and MAE ($p < 0.001$) were significantly lower in the model of the present study than those in the model of Miyashiro et al. [15]. However, the MSEs of both models followed a similar but unique trend, in which MSEs increased up to TD6 and then consistently decreased towards TD10.

Table 2 shows the comparison of regression equations and their evaluation metrics with and without reaction time. The regression equation yielded from the net touchdown times (i.e., without reaction time) improved the goodness of fit only at the first hurdle (TD1 in Table 2).

Table 3 shows the summary of evaluation metrics with unifying the scale of each touchdown time in the present study model. The MSEs were found to decrease linearly from TD1 to TD10 while the unique trend seen before unifying the scale was no longer observed. On the other hand, $R^2$ still kept a trend, which consistently increased up to the last touchdown time as same as before unifying the scale.

## Discussion

The present study aimed to establish a more representative statistical model of touchdown times based on the data of elite 110 m hurdlers to precisely predict their typical touchdown times. Throughout all touchdown times, the proposed predictive equations showed a better prediction performance ($R^2 = 0.527$–$0.981$, MSE $= 0.0015$–$0.0028$, MAE $= 0.019$–$0.033$)

**Table 3. Summary of evaluation metrics with unifying the scale of each touchdown time in the present study model.**

| Touchdown | $R^2$ | | MSE | |
|---|---|---|---|---|
| | Not scaled | Scaled | Not scaled | Scaled |
| TD1 | 0.527 | 0.536 | 0.0015 | 0.4762 |
| TD2 | 0.670 | 0.677 | 0.0018 | 0.3318 |
| TD3 | 0.784 | 0.789 | 0.0021 | 0.2171 |
| TD4 | 0.836 | 0.840 | 0.0025 | 0.1643 |
| TD5 | 0.881 | 0.884 | 0.0027 | 0.1190 |
| TD6 | 0.914 | 0.916 | 0.0028 | 0.0868 |
| TD7 | 0.939 | 0.940 | 0.0027 | 0.0613 |
| TD8 | 0.959 | 0.960 | 0.0023 | 0.0408 |
| TD9 | 0.970 | 0.970 | 0.0022 | 0.0304 |
| TD10 | 0.981 | 0.981 | 0.0017 | 0.0190 |
| Mean (SD) | 0.846 (0.141) | 0.849 (0.138) | 0.0022 (0.0004) | 0.1546 (0.1415) |

TD: touchdown time. $R^2$: adjusted coefficient of determination.

MSE: mean squared error.

compared with those of existing equations made by Miyashiro et al. [15] ($R^2$ = 0.481–0.979, $MSE$ = 0.0017–0.0039, $MAE$ = 0.034–0.063). These findings supported our hypothesis that the model established in this study would show a better generalization performance to elite 110 m hurdlers compared to that of the model of Miyashiro et al. [15]. As the existing model has never been evaluated by cross-validation to avoid overfitting the race outcomes to the regression equations, this is the first study that evaluated the regression equations in an appropriate procedure and established a precise prediction touchdown model of the 110 m hurdles.

In both models, the adjusted coefficient of determination ($R^2$) was relatively lower in the initial phase of the race (from the start to the second touchdown time) and then consistently increased up to the last touchdown time (Table 1). A similar tendency was previously reported by Tsiokanos et al. [5]. They found that decisive points of resultant race time were touchdown times from the third to the tenth hurdle ($r$ = 0.81–0.99), suggesting that there is a linear and foreseeable relationship between the resultant race time and touchdown times after the third hurdle. Therefore, it can be assumed that the touchdown times from the start to the second hurdle include rather high, unpredictable variances than those in the latter part of the race. This might explain relatively lower fitting agreements to the statistical model seen in the first two touchdown times of the race ($R^2$ values in Table 1). Of possible factors affecting this outcome, the reaction time of the start included in these touchdown times is a likely one. To examine the impact of reaction time, net touchdown times were extracted from the data of touchdown times that reported reaction times ($N$ = 81) and yielded the regression equation using the net touchdown times. Indeed, the regression equation yielded from the net touchdown times improved the goodness of fit only at the first hurdle (TD1 in Table 2). The use of the net touchdown times failed to show any improvement in the goodness of fit for the other touchdown times compared to those yielded by gross touchdown times including the reaction time. Therefore, it can be suggested that excluding the reaction time from the regression model only provides a limited benefit on predictive performance depending on the position of hurdles.

Mean squared error ($MSE$) represents the amount of error in statistical models. Throughout the race, the model of the present study showed smaller $MSE$s than those of the model of Miyashiro et al. [15], and the mean value was significantly smaller than that of the model of

Miyashiro et al. [15] (Table 1). However, it is interesting to note that *MSEs* of both models followed a similar but unique trend, in which *MSEs* increased up to TD6 and then consistently decreased towards TD10. In general, the *MSE* of TD1 was expected to be the largest and *MSEs* of the latter TD times would decrease almost linearly towards TD10 because the latter TD times will gradually come to be closer to the resultant race time (prediction outcome). One possible explanation for this unreasonable trend of *MSEs* seen in the present study was the different scales of each TD time. After unifying the scales, *MSEs* were found to decrease linearly from TD1 to TD10 (Table 3) while the unique trend seen before unifying the scale was no longer observed. On the other hand, $R^2$ still kept a reasonable trend, which consistently increased up to the last touchdown time as same as before unifying the scale (Table 3) since $R^2$ is a scale-independent evaluation metric. From these results, therefore, it was confirmed that an unreasonable trend seen in *MSEs* was induced by the different scales of TD times.

In the present study, the generalization of the equations was tested using leave-one-out cross-validation and data from the previous study (the race time corresponding to the world 50th rankings during the last decade [24]) were substituted into both models. Consequently, the model of the present study showed a better generalization than the model of Miyashiro et al. [15] (Table 2). It is known that the regression model is no longer guaranteed when a prediction is made outside the range of observed data [27]. The model of Miyashiro et al. [15] used the data of intermediate-level hurdlers as training data, which probably disturbed its generalization performance for elite hurdlers. Therefore, it can be stressed here that the regression equations proposed in this study would be more appropriate for training interventions of elite 110 m hurdlers than the conventional model. Elite or sub-elite hurdlers and their coaches may achieve their target race times using the model proposed in this study. The model also allows them to have a more concrete race evaluation (actual vs. predicted touchdown times) and intervention plan for subsequent training sessions.

The present study is not without limitations. Firstly, we did not consider the case of hurdlers who hit the hurdles. Hitting the hurdles sometimes happens in the 110 m hurdles race [14], and has been noted to be associated with interval running performance [11, 28, 29]. Further investigation of considering hitting hurdles appears to be warranted. Secondly, we did not take the wind speed during the race into account. Wind speed can affect the race time of the 110 m hurdles race [30]. Moinat et al. [30] reported that 2.0 m/s tailwind provides a mean advantage of 0.146 s for the 110 m hurdles. Thus, considering these factors would change the model performance and it is an interesting future direction. Beyond these limitations, the proposed predictive equations time based on the statistical model would provide helpful information for coaching.

## Supporting information

**S1 Data. The data underlying the findings in this study.**
(CSV)

**S2 Data. Python code.**
(IPYNB)

## Author Contributions

**Conceptualization:** Ryo Iwasaki.

**Data curation:** Ryo Iwasaki.

**Formal analysis:** Ryo Iwasaki, Kento Nozawa.

**Investigation:** Ryo Iwasaki.

**Methodology:** Ryo Iwasaki, Kento Nozawa.

**Project administration:** Ryo Iwasaki.

**Resources:** Ryo Iwasaki.

**Software:** Ryo Iwasaki, Kento Nozawa.

**Supervision:** Hiroyuki Nunome, Kento Nozawa.

**Validation:** Ryo Iwasaki, Kento Nozawa.

**Visualization:** Ryo Iwasaki.

**Writing – original draft:** Ryo Iwasaki.

**Writing – review & editing:** Ryo Iwasaki, Hiroyuki Nunome, Kento Nozawa.

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
