## [Decision Letter · Decision Letter 0]

28 Sep 2022

PONE-D-22-21310New predictive model of the touchdown times in a high level 110-m hurdlesPLOS ONE

Dear Dr. Iwasaki,

Thank you for submitting your manuscript to PLOS ONE. After careful consideration, we feel that it has merit but does not fully meet PLOS ONE’s publication criteria as it currently stands. Therefore, we invite you to submit a revised version of the manuscript that addresses the points raised during the review process.

We look forward to receiving your revised manuscript.

Kind regards,

Laura-Anne Marie Furlong

Academic Editor

PLOS ONE

Journal Requirements:

Reviewers' comments:

Reviewer's Responses to Questions

**Comments to the Author**

1. Is the manuscript technically sound, and do the data support the conclusions?

Reviewer #1: Yes

Reviewer #2: Partly

2. Has the statistical analysis been performed appropriately and rigorously? 

Reviewer #1: Yes

Reviewer #2: Yes

3. Have the authors made all data underlying the findings in their manuscript fully available?

Reviewer #1: Yes

Reviewer #2: Yes

4. Is the manuscript presented in an intelligible fashion and written in standard English?

Reviewer #1: No

Reviewer #2: Yes

5. Review Comments to the Author

Reviewer #1: The aim of this study was to establish a novel statistical model of touchdown times in the men’s high hurdles. The paper is sound in most ways but there isn’t a strong rationale for the study (the main one seems to be to compare to another study) and the discussion doesn’t provide enough information that makes the findings practically useful for a coach. It would be beneficial to obtain the services of a native speaker with expertise in hurdling research who can help rewrite your work in idiomatic English.

Line 19 – “top-level” is not a good description to use (for one thing, “standard” is a better word than “level”).

Lines 20/21, 63, 90, etc. – please use the correct SI unit for seconds, which is ‘s’.

Lines 31-32 – your abstract has not explained well what the point of the study is or what value it is to coaches. Could you please provide some tangible information that a coach could take away from your study?

Line 36 – it is a bit odd that you have reported the height of the hurdles in cm, but the distance in m (please use m for all heights and distances, as it is the SI unit). Also, please place a space between the numeral and the unit, as per standard practice.

Line 38 – what is a “high-level hurdle clearance technique”? Is there a definition of this? Could you refer to previous literature on world-class hurdlers to provide some information?

Line 39 – what is an “interval” defined as?

Lines 41-42 – you should replace “top-level” and “lower-level” with “better” and “worse” (or even “faster” and “slower”).

Line 45 – I would argue that it is not “most likely predetermined” but definitely predetermined. It might be useful to refer to previous literature on hurdling to define the three intermediate steps.

Line 50 – a concluding sentence here that explains how this links with your study would be useful.

Line 57 – this sentence is not well written, so please rephrase it.

Line 60 – I am not sure you should have the hyphen in this sentence.

Line 65 – the fact that it was published in Japanese is unimportant. Please remove this point.

Line 70 – please replace “unseen” with “excluded” or something similar.

Lines 76-77 – if you want to provide useful information for non-elite hurdlers, why do you include data from elite hurdlers?

Line 89 – the correct name for these championships was (until recently) the “IAAF World Championships”.

Line 119, 141, 205, etc. – please change “average” to “mean” (as I assume this is what you are referring to).

Line 135 – you have used US English spellings in other parts of the paper, so to be consistent please change “grey” to “gray”.

Line 156 – please change “compared to” to “compared with”.

Line 180 onwards – shouldn’t most of this information be included in the Results section?

Lines 216-232 – shouldn’t most of this information be included in the Methods section?

Lines 251-261 – although most of what is in your discussion seems scientifically sound, you haven’t provided any useful information on hurdling for coaches or athletes. For example, how are these data useful for developing coaching regimens? What do they tell us about improving performance? At the moment, it seems that you have simply found that those athletes who reach certain hurdles quicker have higher finishing positions. This is not particularly interesting for coaches.

Lines 304-309 – where you have written “Brian T”, I assume you mean “Gearity, BT”. Please check all of your references. I would also recommend you read some of the latest research on elite hurdling, which will help with your introduction in particular, such as:

Bissas, A., Paradisis, G. P., Hanley, B., Merlino, S. & Walker, J. (2022). Kinematic and temporal differences between World-class men’s and women’s hurdling techniques. Frontiers in Sports and Active Living, 4, 873547.

Hanley, B., Walker, J., Paradisis, G. P., Merlino, S. & Bissas, A. (2021). Biomechanics of world-class men and women hurdlers. Frontiers in Sports and Active Living, 3, 704308.

Reviewer #2: GENERAL COMMENTS:

This paper presents a revised prediction model for the 110 m sprint hurdles event based on a cross validated regression model. The paper is technically sound and the conclusions are supported by the data. The writing is generally clear and intelligible, however the structure of the paper does not adhere to reporting guidelines and major revisions are required in restructuring. In particular, results appear in the discussion and method related information appears in the in the results. These shortcomings can be rectified but will require extensive revision. Overall the topic has merit and will be of value to sports scientists and practitioners. The authors repeatedly claim that this is a novel method of analysis however, LOOCV is a well established technique except that it has not been applied to these data. I advise the claim of novel methods should be omitted as this is not accurate and it will not detract from the originality of the study to remove the claim of novel methods.

SPECIFIC COMMENTS:

P 2 ln 18: I don't think the methods used in this model are novel. The regression and cross validation techniques used in this study has been used for many years.

P 2 ln 19: to predict performance based on touchdown times.

P2 ln R2 should be R (2 superscript)

P3 ln 39: "time of interval running" verbose and unclear. Hurdle touchdown period or intervals is easier to understand

P3 ln 44: There are various verbose expression that appear throughout for example "to make the interval running time shorter" ... perhaps " to reduce touchdown periods". Please do a thorough recheck and revise.

P3 Ln 46: Spatio temporal variables is vague, perhaps just be specific ... you only used hurdle touchdown times.

P3 ln 60: "To the best of our knowledge" perhaps avoid this expression ... "The model appears to be based only on...."

P4 ln 66: "it is quite unlikely to apply the model to predict temporal parameters" poor grammar, please revise.

P4 ln 67 "evaluated the prediction accuracy of the regression model using the training data", Training data in the context of the study could be confusing here since training data could be data derived out of competition or it refers to training data sets for the regression model. please try to be clear.

P4 ln 84: Please clarify whether these data were based only on competition data and if not why data from a training situation were not removed.

P8 ln 152: "novel statistical model" this is not a novel approach

P8 ln 155 "better predictable performance" perhaps Better prediction performance"

P9 ln 175-185: this information is out of sequence as it is methods information. Please follow standard structure

P9 ln 178: This is a misuse of correlation and significance. The fact that a correlation is not statistically significant does not mean that RT had no effect. To illustrate this if two samples are compared and the t test finds no statistically significant difference does not mean that the mean scores are equal. Therefore you cannot dismiss the impact of RT on performance by reporting that the correlation of RT with final performance in not significant. Later this idea is rectified but you should not imply the lack of significance means no difference or no effect.

P10 ln 196

This table presents results and should appear in the results section not the discussion

P11 ln 216-221: this is Methods

P12 ln 234: The table presents results and should be in the results section.

Major restructuring of the results and discussion sections are required.

6. PLOS authors have the option to publish the peer review history of their article (what does this mean?). If published, this will include your full peer review and any attached files.

Reviewer #1: No

Reviewer #2: **Yes: **Andrew J Harrison

---

## [Author Response · Author response to Decision Letter 0]

16 Oct 2022

We have submitted our revision paper and response to reviewers.

Sincerely yours,

---

## [Decision Letter · Decision Letter 1]

6 Nov 2022

PONE-D-22-21310R1New predictive model of the touchdown times in a high level 110-m hurdlesPLOS ONE

Dear Dr. Iwasaki,

Thank you for submitting your manuscript to PLOS ONE. After careful consideration, we feel that it has merit but does not fully meet PLOS ONE’s publication criteria as it currently stands. Therefore, we invite you to submit a revised version of the manuscript that addresses the points raised during the review process.

We look forward to receiving your revised manuscript.

Kind regards,

Laura-Anne Marie Furlong

Academic Editor

PLOS ONE

Journal Requirements:

Reviewers' comments:

Reviewer's Responses to Questions

**Comments to the Author**

1. If the authors have adequately addressed your comments raised in a previous round of review and you feel that this manuscript is now acceptable for publication, you may indicate that here to bypass the “Comments to the Author” section, enter your conflict of interest statement in the “Confidential to Editor” section, and submit your "Accept" recommendation.

Reviewer #1: All comments have been addressed

Reviewer #2: (No Response)

2. Is the manuscript technically sound, and do the data support the conclusions?

Reviewer #1: Yes

Reviewer #2: Yes

3. Has the statistical analysis been performed appropriately and rigorously? 

Reviewer #1: Yes

Reviewer #2: Yes

4. Have the authors made all data underlying the findings in their manuscript fully available?

Reviewer #1: Yes

Reviewer #2: Yes

5. Is the manuscript presented in an intelligible fashion and written in standard English?

Reviewer #1: Yes

Reviewer #2: No

6. Review Comments to the Author

Reviewer #1: Thank you for addressing my comments. Your manuscript is very much improved, and I congratulate you on your work.

Reviewer #2: Thank you for your revised manuscript. In general the revisions have addressed the technical issues I identified in the previous review, however there remain several grammatical and expression issues throughout the revised paper. Some of these grammatical errors appear in the revised sections while other errors appear in the previous text. I have included details of specific revisions required but a detailed proof check of the manuscript is still required.

Specific comments:

Ln 20 : perhaps elite rather than elite-standard throughout. Also you need to be consistent in the format of number unit especially with respect to distances we have 110-m, 1.067 m appearing best to have one format consistently I suggest 110 m not 110-m

Ln 33-37: suggest you use “this study” rather than "the present study”

Ln 36: “….would help to improve training interventions…..”

Ln 44: replace “namely” with “by”

Ln 46: reconsider: significantly correlated (r = 0.81–0.99); perhaps, “strongly correlated (r = 0.81–0.99; p<0.05)”

Ln53-54: Are spatiotemporal variables important determinants of performance rather than just “important for better performance”?

Ln 59: replace “To do so, this is necessary“ with “this is necessary”.

Ln 66-67: the grammar in this sentence is poor and therefore expression is unclear. Please revise.

Ln 73-75: “Thus, it is reasonable to interpret that the existing model is inapplicable to predicting…” this expression is verbose, please simplify (I suggest: “The existing model therefore fails to reliably predict the touchdown times…..”)

Ln 75-77: Moreover…..prediction accuracy”. This new sentence is poorly constructed and it does not add anything of value to the text, I suggest you omit it.

Ln 83 – 86: In my earlier review, I suggested the need to strengthen the rationale for the study or make the rationale more obvious. This can be done by simply stating that based on the previous discussion, it is clear that the existing model does not provide accurate predictions, therefore a more robust and reliable model is required. Your text in this section is verbose and misses this simple statement of need.

Ln 97 omit "actual"

Ln 137-138 try “ a numerical experiment for net touchdown times (without the reaction time) was completed.

Ln 139: try “touchdown times with reported reaction times”

Ln 233 omit possess

Ln 268: revise : “The model also allows…”

7. PLOS authors have the option to publish the peer review history of their article (what does this mean?). If published, this will include your full peer review and any attached files.

Reviewer #1: No

Reviewer #2: **Yes: **Andrew J Harrison

---

## [Author Response · Author response to Decision Letter 1]

17 Nov 2022

We wish to express our strong appreciation to the reviewer for comments on our paper. We feel the comments have helped us significantly improve the paper.

---

## [Decision Letter · Decision Letter 2]

22 Nov 2022

New predictive model of the touchdown times in a high level 110 m hurdles

PONE-D-22-21310R2

Dear Dr. Iwasaki,

We’re pleased to inform you that your manuscript has been judged scientifically suitable for publication and will be formally accepted for publication once it meets all outstanding technical requirements.

Kind regards,

Ryan Thomas Roemmich

Academic Editor

PLOS ONE

Additional Editor Comments (optional):

Reviewers' comments:

Reviewer's Responses to Questions

**Comments to the Author**

1. If the authors have adequately addressed your comments raised in a previous round of review and you feel that this manuscript is now acceptable for publication, you may indicate that here to bypass the “Comments to the Author” section, enter your conflict of interest statement in the “Confidential to Editor” section, and submit your "Accept" recommendation.

Reviewer #1: All comments have been addressed

Reviewer #2: All comments have been addressed

2. Is the manuscript technically sound, and do the data support the conclusions?

Reviewer #1: Yes

Reviewer #2: Yes

3. Has the statistical analysis been performed appropriately and rigorously? 

Reviewer #1: Yes

Reviewer #2: Yes

4. Have the authors made all data underlying the findings in their manuscript fully available?

Reviewer #1: Yes

Reviewer #2: Yes

5. Is the manuscript presented in an intelligible fashion and written in standard English?

Reviewer #1: Yes

Reviewer #2: Yes

6. Review Comments to the Author

Reviewer #1: Thank you again for addressing my comments. As stated before, I congratulate you on your work and look forward to seeing it in print.

Reviewer #2: All my comments have been satisfactorily address. My congratulation to the the authors on their work.

7. PLOS authors have the option to publish the peer review history of their article (what does this mean?). If published, this will include your full peer review and any attached files.

Reviewer #1: No

Reviewer #2: **Yes: **Andrew J Harrison

---

## [Editor Report · Acceptance letter]

25 Nov 2022

PONE-D-22-21310R2 

New predictive model of the touchdown times in a high level 110 m hurdles 

Dear Dr. Iwasaki:

I'm pleased to inform you that your manuscript has been deemed suitable for publication in PLOS ONE. Congratulations! Your manuscript is now with our production department. 

Kind regards, 

on behalf of

Dr. Ryan Thomas Roemmich 

Academic Editor

PLOS ONE